Figure 1: Demonstration of the proposed method.

# PRIOR-BASED 4D HUMAN-SCENE RECONSTRUCTION FROM MONOCULAR VIDEOS

## ABSTRACT

Accurately capturing dynamic humans as they interact with their 3D environment from a single camera is a pivotal goal for applications spanning from assistive robotics to AR. However, current monocular approaches fall short, as they are typically restricted to reconstructing either the person or the static background in isolation. Methods that capture both often rely on cumbersome multi-view setups, limiting their real-world applicability. To this end, we propose a novel framework that reconstructs the complete 4D human-scene representation from monocular video. We formulate the task as an ill-posed inverse problem and introduce a robust regularization strategy that leverages two complementary priors: a static 3D Gaussian Splatting representation of the scene, and an animatable, SMPL-based 3D Gaussian avatar of the human. Our method jointly optimizes the camera pose, human motion, and the parameters of both priors to faithfully reconstruct time-varying geometry, appearance, and physically plausible human-scene interactions. We validate our approach on a self-collected dataset featuring synchronized human acting videos, human and scene scan videos. Our results demonstrate state-of-the-art performance, achieving average 23 dB PSNR on challenging novel views and surpassing existing monocular baselines.

## 1 INTRODUCTION

From autonomous service robots and AR navigation to telepresence and sports broadcasting, many everyday applications reply on perceiving human activities in context. Effective interaction requires not only recognizing what people are doing, but also understanding the 3D layout of the surrounding environment and how people move and make contact with it over time. This motivates 4D human-scene reconstruction, jointly recovering the time-varying geometry, appearance, and motion of humans and scenes, along with their spatial relationships. To be practical and cost-effective beyond controlled studios, the method should generalize across individuals, clothing, and environments, and operate on monocular RGB videos captured with smartphones. However, existing approaches often focus only on the human Qian et al. (2024); Hu et al. (2024b) or the static scene Xu et al. (2024a); Wen et al. (2025), or assume multi-view synchronized cameras and extensive calibration Xu et al. (2024b); Wang et al. (2025), limiting applicability in real-world settings.

Although existing monocular 4D human-scene reconstruction method Zhang et al. (2025); Kim et al. (2025); Moon et al. (2024a) are successful in their respective tasks, the challenge remains in generalizing to novel viewpoints outside the input camera trajectory. Some artifacts may be produced, such as the elongated 3D Gaussian floaters in scenes or the opacity leakage and wrong SMPL pose of humans. It's caused by the fact that the monocular 4D human-scene reconstruction problem is naturally ill-posed with multiple valid solutions, which means even if the rendered results appear realistic at the training view, they can still correspond to the inaccurate spatial geometry and appearance features of the avatar and scene representations. Consequently, there is a growing need for methods that reconstruct humans and scenes from monocular videos and deliver faithful novel view rendering in everyday scenarios.

Following the approach to address ill-posed problems by mathematicians Bertero et al. (1988), we propose a novel method that integrates scene and human priors as regularization to transform the ill-posed problem into well-posed approximations. By incorporating priors, the optimization of the canonical avatar and scene becomes relatively tractable, and the complex problem of modeling point motions is reformulated as estimating SMPL parameters along with the corresponding deformations.

As shown in Fig. 2, our framework first extracts two complementary priors: a static scene prior from a scene-scan video using 3D Gaussian Splatting Kerbl et al. (2023), and an animatable human prior from a human-scan video via an SMPL-based 3D Gaussian avatar reconstruction model Moon et al. (2024a). These priors serve as regularization, enabling a monocular acting video to be used for camera tracking, human pose estimation, and dense photorealistic reconstruction of both the scene and the human. However, directly combining these priors presents three challenges: **First**, monocular reconstructions suffer from scale ambiguity, leading to mismatched scales between the avatar and the scene. Naive integration often results in floating humans or penetrations; **Second**, clothing is inherently deformable, and pose-dependent non-rigid deformations such as folds cannot be captured by a fixed avatar model; **Third**, since all three input videos are recorded on smartphones, appearance discrepancies arise between the priors and the acting video, producing color inconsistencies in reconstruction which may destroy the geometric consistency.

To address these issues, we design three targeted solutions. For the first challenge, we introduce an optimizable parameter $\sigma$ to adjust avatar size, and a contact loss to enforce spatial contact between human and scene. For the second challenge, we augment the SMPL avatar with pose-dependent offsets and incorporate a normal loss to better capture clothing related deformations. For the third challenge, we optimize only the appearance features of the scene, together with the captured camera poses, preventing overfitting of geometry to training views and preserving geometric consistency. For training and evaluation, We construct a dataset that includes six static scenes, four human scan sequences, and six acting videos, with three additional sequences exclusively for evaluation. On novel viewpoint rendering, our approach attains an average PSNR of 23.5 dB, outperforming the baseline by 3 dB.

In summary, our contributions are:

- A novel method is proposed to solve the novel view synthesis challenges of 4D human-scene reconstruction in the wild by introducing scene and avatar priors.
- We refine the priors through the 2D/3D knowledge extracted from the monocular acting video.
- We collect a real-world dataset consisting of acting videos paired with additional human and scene scans. The effectiveness of our method is evaluated by extensive ablation studies and comparisons.

## 2 RELATED WORKS

### 2.1 DYNAMIC RECONSTRUCTION

To reconstruct dynamic scenes, several works, such as D-NeRF Pumarola et al. (2021) and HyperReel Attal et al. (2023), leverage implicit representations to model time-varying radiance fields but suffer from high computational costs and slow convergence. Voxel-based approaches like Hex-Plane Cao & Johnson (2023) and V4D Gan et al. (2023) improve efficiency through 4D grid factorization, yet they struggle to balance resolution with memory constraints. In contrast, 3D Gaussian Splatting Kerbl et al. (2023) offers real-time rendering and explicit geometric control, making it a promising candidate for dynamic extensions. Building on this momentum, several works have

sought to extend Gaussian-based methods to model temporal dynamics. Luiten et al. Luiten et al. (2024) introduced a foundational framework for persistent dynamic view synthesis by tracking 3D Gaussians over time. Concurrently, Yang et al. Yang et al. (2024b), Wu et al. Wu et al. (2024b), and Liang et al. Liang et al. (2023) proposed joint optimization strategies that couple canonical-space Gaussians with deformation fields to disentangle geometry and motion. These approaches implicitly assume topological consistency by anchoring dynamics to a static canonical configuration, limiting their applicability to scenes with rigid or semi-rigid motions. Kratimenos et al. Kratimenos et al. (2024) further decomposed motion into neural trajectories to enforce locality and rigidity priors, enhancing robustness for monocular video inputs. While in various dynamic scenarios, these methods achieve compelling rendering results on viewpoints that lie within the training trajectory, they can struggle with maintaining geometric consistency when rendered under out-of-distribution poses.

## 2.2 Animatable 3D Human Avatars

The creation of animatable 3D human avatars has advanced significantly through diverse 3D representations and learning frameworks. Early works leveraged parametric meshes, such as SMPL, extended via per-vertex offsets Alldieck et al. (2018) or driven by conditional variational autoencoders Bagautdinov et al. (2021). Localized representations, like texel-aligned features Remelli et al. (2022), improved detail preservation. The advent of neural radiance fields (NeRF) Mildenhall et al. (2021) spurred volumetric approaches, enabling high-fidelity avatars from multi-view studio captures Peng et al. (2021a;b); Shen et al. (2023). Concurrently, methods like Kwon et al. Kwon et al. (2021) incorporated vertex-aligned features for enhanced generalization. Recent efforts prioritize monocular video inputs, circumventing the need for 3D supervision (e.g., Jiang et al. Jiang et al. (2022), Guo et al. Guo et al. (2023), and Jiang et al. Jiang et al. (2023)). The rise of 3D Gaussian Splatting (3DGS) Kerbl et al. (2023) further accelerated progress, enabling efficient rendering in avatars via triplane decomposition Kocabas et al. (2024), SMPL-aligned positional maps Hu et al. (2024a), or expressive Gaussian-based models Liu et al. (2024). Despite these advances, most methods focus on body motion, neglecting hand and facial animation. While X-Avatar Shen et al. (2023) supports whole-body animation, it requires extensive 3D-registered facial data and struggles with in-the-wild monocular videos. Similarly, Liu et al. Liu et al. (2024) lack facial expression control. Parametric models like SMPL-X Pavlakos et al. (2019) unify body, face, and hands, inspiring pose estimation methods Choutas et al. (2020); Moon et al. (2022), yet translating these to expressive, lightweight avatars remains challenging. Hand-specific models, such as UHM Moon et al. (2024b) and its relightable extension Chen et al. (2024), highlight the complexity of part-specific modeling, underscoring the need for holistic, practical solutions. However, these methods generally focus only on the reconstruction of an animatable avatar in the canonical space alone, with the static scene being largely overlooked.

## 2.3 Human-Scene Reconstruction

The field of human-scene reconstruction has rapidly progressed from independent reconstructions of humans and environments toward unified frameworks that jointly recover dynamic human motion and surrounding scenes with neural representation Kim et al. (2025); Xue et al. (2024); Zhang et al. (2025); Moon et al. (2024a); Liu et al. (2025). JOSH Liu et al. (2025) initializes with local scene reconstruction and human mesh recovery, and subsequently performs joint optimization of human motion and scene geometry under human–scene contact constraints. Showmak3r Kim et al. (2025) reconstructs dynamic radiance fields from TV shows, facilitating scene editing analogous to operations in a production control room. Furthermore, the method incorporates Score Distillation Sampling Loss Lee et al. (2024) to alleviate artifacts that typically appear in novel viewpoints. ODHSR Zhang et al. (2025) leverages monocular RGB videos of humans to jointly reconstruct a dense photorealistic Gaussian representation of both the scene and the dynamic human, and simultaneously recover camera poses, human motion, and human silhouettes under a SLAM formulation. HSR Xue et al. (2024) and Exavatar Moon et al. (2024a) are the most closely related works to ours. HSR jointly optimizes dynamic human and scene geometry using neural implicit representations, with supervision provided by monocular depth and surface normals estimated from full-frame predictions. However, it often fails to faithfully capture human motion from novel viewpoints. Similar to our approach, Exavatar leverages 3D Gaussians to represent both the full-body human avatar and the surrounding scene. Nevertheless, it often produces geometric inconsistencies in the scene and suffers from opacity leakage artifacts in the avatar.

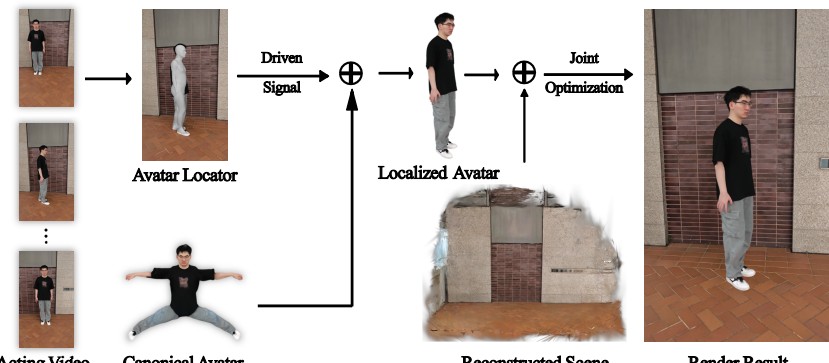

Figure 2: This is the pipeline of our method. Given an acting video, this model first extract the SMPL-X parameter used to drive and localize the avatar prior in the scene prior. Then we perform photometric, normal loss as 2D supervision, contact and regularization loss as 3D supervision to optimize jointly the camera pose, human motion, and the parameters of both priors. After optimization, the output representation can achieve realistic novel view synthesis.

## 3 PRELIMINARIES

Prior-based 4D human-scene Gaussian reconstruction involves both static scene and dynamic human reconstruction. We introduce preliminaries of these two tasks in Sec.3.1 and Sec.3.2, respectively.

### 3.1 STATIC SCENE PRIOR RECONSTRUCTION(GS SCENE)

Our goal is to obtain a dense reconstruction of the static background from a scene–scan video. Following 3D Gaussian Splatting (3DGS) Kerbl et al. (2023), we explicitly represent the background as a set of $M_s$ anisotropic Gaussians in the world frame $\mathcal{G}_b = \left\{ (\mathbf{V}_{b,i}, \alpha_{b,i}, \mathbf{\Sigma}_{b,i}, \mathbf{z}_{b,i}) \right\}_{i=1}^{M_s}$, where the center $\mathbf{V}_{b,i} \in \mathbb{R}^3$, opacity $\alpha_{b,i} \in [0, 1]$, covariance $\mathbf{\Sigma}_{b,i} \in \mathbb{R}^{3\times3}$, and SH appearance coefficients $\mathbf{z}_{b,i}$ parameterize geometry and color.

$$\mathbf{\Sigma}_{b,i} = \mathbf{R}_{b,i}\, \mathbf{S}_{b,i}\, \mathbf{S}_{b,i}^\top\, \mathbf{R}_{b,i}^\top, \quad \mathbf{R}_{b,i} \in SO(3),\ \mathbf{S}_{b,i} = \mathrm{diag}(s_x, s_y, s_z). \tag{1}$$

Appearance is modeled with spherical harmonics (SH) to achieve view–dependent color. To obtain reliable geometry and mitigate floating–Gaussian artifacts, we supervise the rendered depth with monocular depth from Depth-Anything v2 Yang et al. (2024a) and extend the 3DGS objective with regularizers that improve surface consistency Kerbl et al. (2024). Therefore, we only optimize its SH parameters $\mathbf{z}_b$ in the subsequent optimization process, as described in Sec.1.

### 3.2 ANIMATABLE HUMAN AVATAR

Our goal is to obtain a fully drivable human avatar model from a single-person scan video. To faithfully capture fine details and achieve photorealistic rendering, the avatar must support whole-body drivability, including the body, hands, and facial expressions. For this purpose, we adopt the off-the-shelf method ExAvatar Moon et al. (2024a) for the Gaussian avatar reconstruction.

Unlike conventional Gaussian representations that use ellipsoids with opacity and spherical har-monic (SH) coefficients, the formulation is simplified by representing each Gaussian as a solid sphere with direct RGB values for appearance to avoid serious overfitting. In the canoni-cal space, the neutral posed avatar is parameterized as a set of optimizable variables $\mathcal{G}_h = \left\{ \boldsymbol{\beta}, \sigma, \left\{ (\mathbf{V}_{h,i}, \mathbf{S}_{h,i}, \mathbf{C}_{h,i}) \right\}_{i=1}^{M_h} \right\}$, where shape vector $\boldsymbol{\beta} \in \mathbb{R}^{100}$, global scale $\sigma \in \mathbb{R}$, 3D position $\mathbf{V}_{h,i} \in \mathbb{R}^3$, scale $\mathbf{S}_{h,i} \in \mathbb{R}$, and RGB color $\mathbf{C}_{h,i} \in \mathbb{R}^3$.

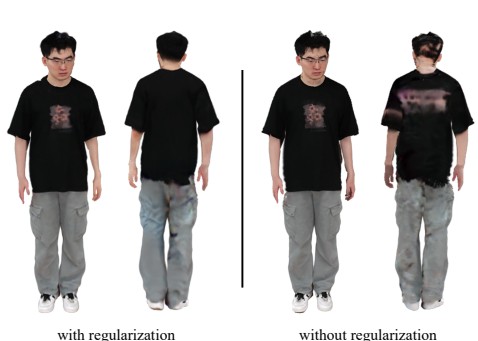

with regularization    without regularization

Figure 3: Qualitative ablation of Regularization Loss. The regularization significantly reduces artifacts at both train and novel views.

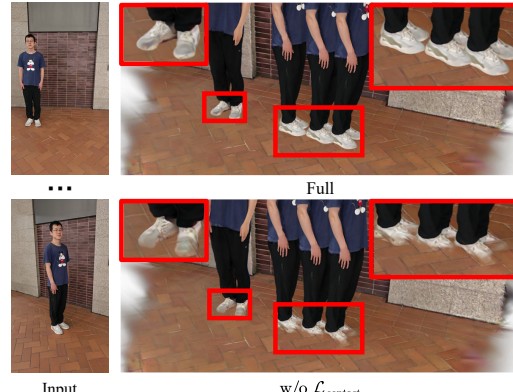

Figure 4: Qualitative ablation of $\mathcal{L}_{\text{contact}}$. The corner inset illustrates the human–scene contact situation.

## 4 METHOD

In this section, we first formalizes the overall 4D human-scene reconstruction framework in Sec. 4.1. Then the initialization process is proposed in Sec. 4.2. After that, we address how to accurately deform the avatar and localize the avatar within the scene in Sec. 4.3. Finally, we describe the optimization method in Sec. 4.4.

### 4.1 PROBLEM SETUP

To solve the 4D human-scene Gaussian reconstruction problem, we need to reconstruct three key components: the static scene, animatable avatar and whole-body driving signals. The inputs consist of two monocular scanning videos for constructing the scene and human prior respectively, and an acting video for human-scene joint reconstruction. The output is a Gaussian representation of the static scene $\mathcal{G}_b$, and a conanational Gaussian asset of the animatable avatar $\mathcal{G}_h$ with the SMPL-X driving signals $\left\{ E^t, P_c^t, \theta^t, \phi^t \right\}_{t=0}^{M}$, where $\theta$ represents the 3D joint poses of the SMPL-X model, $\phi$ denotes the facial expression parameters, and M is the total number of frames in the acting video.

### 4.2 INITIALIZATION

From the acting video, we extract frames and apply off-the-shelf models to regress SMPL-X and FLAME parameters, as well as 2D keypoint annotations Contributors (2020). These are then jointly optimized to achieve alignment with the detected keypoints.

To further ground the avatar in the scene, we extract 3D ground points from the scene prior with masks predicted by the Segment Anything Model Ren et al. (2024) and fit a plane $\pi : \mathbf{n}^\top x + d = 0$ using RANSAC. The estimated plane provides the initial global scale factor $\sigma$ for human–scene contact. The final output is the optimized SMPL-X parameter set is $\left\{ \beta, \sigma, \left\{ P_c^t, \theta^t, \phi^t \right\}_{t=0}^{M} \right\}$, where $\beta$ is the shape parameter of the SMPL model. Camera poses $E^t$ in the scene prior's coordinate system are estimated by segmenting the human foreground $M^t$ with Segment Anything Ren et al. (2024) and applying structure-from-motion (SfM) Sarlin et al. (2019) jointly to obtain both a sparse scene point cloud and initial acting camera poses. For supervising fine-grained surface geometry, especially clothing deformations, we employ the human surface normal foundation model Khirodkar et al. (2024) to generate per-frame normal maps from the acting video, which are used as ground-truth normals $N$ during optimization.

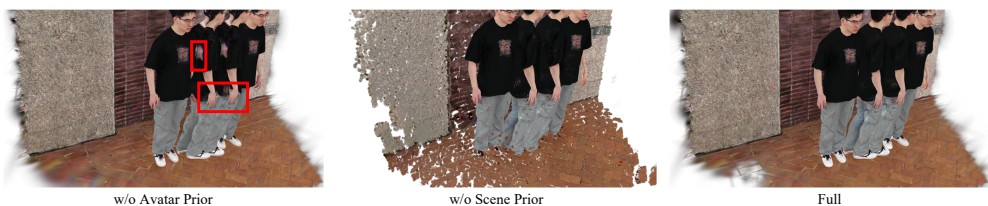

Figure 5: Qualitative ablation of avatar and scene priors.

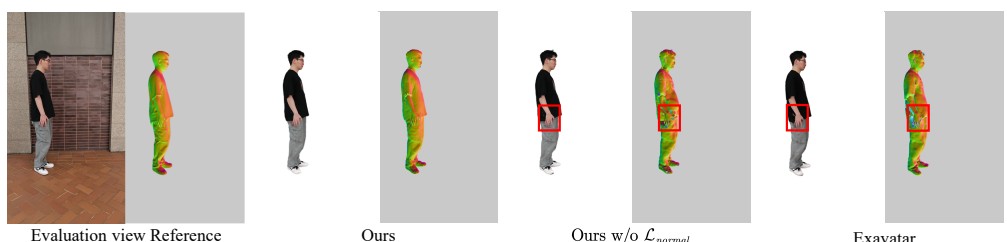

Figure 6: Qualitative ablation study of $\mathcal{L}_{\text{normal}}$ from a novel evaluation view rendering of the avatar, with comparisons against the baseline method.

### 4.3 AVATAR LOCATOR AND RENDERING

#### 4.3.1 AVATAR DEFORMATION AND LOCALIZATION

To model the deformation and movements of human, we drive the avatar's 3D Gaussian assets from canonical space to posed space using SMPL-X signals. Given the neutral posed vertex positions $\mathbf{V_h} = \mathbf{V_h}(\beta)$, we first apply the per-frame expression code $\phi^t$ to produce expression dependent offsets on facial vertices. To capture clothing in the canonical space, we introduce an id-dependent cloth offset $\Delta\mathbf{V_{cloth}}$ specific for avatar, and a pose-dependent offset $\Delta\mathbf{V_{pose}}(\theta^t)$ that models deformation. The similar operation is applied to the scaling matrices $S_h$. The vertices in the camera coordinate are then obtained via linear blend skinning (LBS) under pose $\theta^t$:

$$\bar{\mathbf{V}}_h^t = \mathbf{V}_h + \Delta\mathbf{V}_{\text{expr}}(\phi^t) + \Delta\mathbf{V}_{\text{cloth}} + \Delta\mathbf{V}_{\text{pose}}(\theta^t). \tag{2}$$

$$\mathbf{V}_{h,\text{cam}}^t = \text{LBS}\left(\bar{\mathbf{V}}_h^t, \theta^t, \mathbf{W}\right), \tag{3}$$

Finally, we transform the avatar from the camera coordinate system to the world using the camera extrinsic parameters $\mathbf{V}_{h,\text{world}}^t = E^t * \sigma * \mathbf{V}_{h,\text{cam}}^t$, enabling composition with the static scene.

#### 4.3.2 RENDERING

The rendering of the model at a specified camera pose is achieved using the 3D Gaussian splatting pipeline Kerbl et al. (2023):

$$\hat{I} = \sum_{k=1}^{K} T^k \alpha^k c^k, \quad \text{where } T^k = \prod_{j=1}^{k-1}(1 - \alpha_j) \tag{4}$$

where $c$ represents the RGB color value of each vertex. Following Liang et al. (2024), we also perform volume rendering on the avatar's normal vectors to obtain a 2D normal map for each frame. The spatial normals are computed from the mesh structure defined by the supplied faces and the vertex positions in the camera coordinate system.

$$\hat{N} = \sum_{k=1}^{K} T^k \alpha^k n^k, \quad \text{where } n^k = Normal(\mathbf{V_{h,cam}^t}, F) \tag{5}$$

where $F$ represents the mesh face of SMPL-X human model.

### 4.4 Optimization

#### 4.4.1 Photometric loss

To achieve more realistic rendering results, we employed widely used 2D supervision metrics in computer vision, including the L1 loss of RGB values, D-SSIM and LPIPS These loss functions are separately applied to the Gaussian representations of both avatar and scene, using masks $M^t$ to provide acting video segmentation as the ground truth.

#### 4.4.2 Normal loss

In 4D reconstruction, pure photometric supervision would encourage solutions that ignore geometry, according to the fact that distinct motions can produce identical renderings. It will trap the optimization in spurious local minima. Based on this insight, we exclusively leverage normal maps to supervise the vertex positions of the avatar after deformation and LBS, ensuring the reconstructed avatar has both accurate surface details and consistent geometric structure. Inspired by Xue et al. (2024), we employ a linear combination of L1 and cosine losses of normal rendering, which together define the overall normal loss.

$$\mathcal{L}_{\text{normal}} = \left\| \hat{N} - N \right\|_1 + \left\| 1 - \hat{N}^\top N \right\|_1 \tag{6}$$

#### 4.4.3 Contact loss

For clarity, this work considers a simplified but commonly encountered scenario where the avatar's feet are assumed to remain in contact with the ground. Therefore, avatar's contact points are constrained to the vertices bound at the foot joint. We compute the distances from all foot vertices to the ground plane $\pi$ and select the closest $N_c = 25$ points as contact candidates $\mathbf{V_{contact}}$. To prevent the avatar from either floating above the scene or penetrating the ground, we introduce the contact terms:

$$\mathcal{L}_{\text{contact}} = \sum_{\mathbf{V_{contact}}} \max\big(0,\ |\mathbf{n}^\top \mathbf{V_{contact}} + d| - \Delta_c\big). \tag{7}$$

where $\Delta_c = 0.02$. As describe in Sec.4.2, we has estimated the plane model $\pi$, which can be held constant during optimization as the scene geometry remains fixed throughout the optimization.

#### 4.4.4 Regularization loss

Since the captured views are limited, some body regions would never be observed. To prevent the avatar from overfitting to the observed views and corrupting unseen areas, we directly regularize the avatar's 3D assets.

Unlike traditional avatar methods that rely only on a naked-body mesh prior, we have a high-fidelity scanned avatar prior. Therefore, we introduce the Huber loss that minimizes the difference between the optimized 3D human Gaussians and corresponding prior in both color and scale.

With this formulation, the loss could naturally prevent unreasonable changes in unseen regions, thereby reducing artifacts in views outside the training cameras. As illustrated in Fig. 3, without this loss, the colors of unseen regions, such as the back and the palm become corrupted after the optimization. In contrast, when the loss is applied, the optimized model is able to maintain plausible colors even in unseen regions. Moreover, in the training views, the absence of this loss leads to opacity leakage in regions such as the neck and face.

## 5 Experiments

### 5.1 Setup

#### 5.1.1 Datasets

Our method requires human and scene priors captured from scanned videos. However, to the best of our knowledge, no suitable datasets are publicly available. Therefore, we recorded six sequences by ourselves, including six different static scenes, four human scanning videos, and six acting videos.

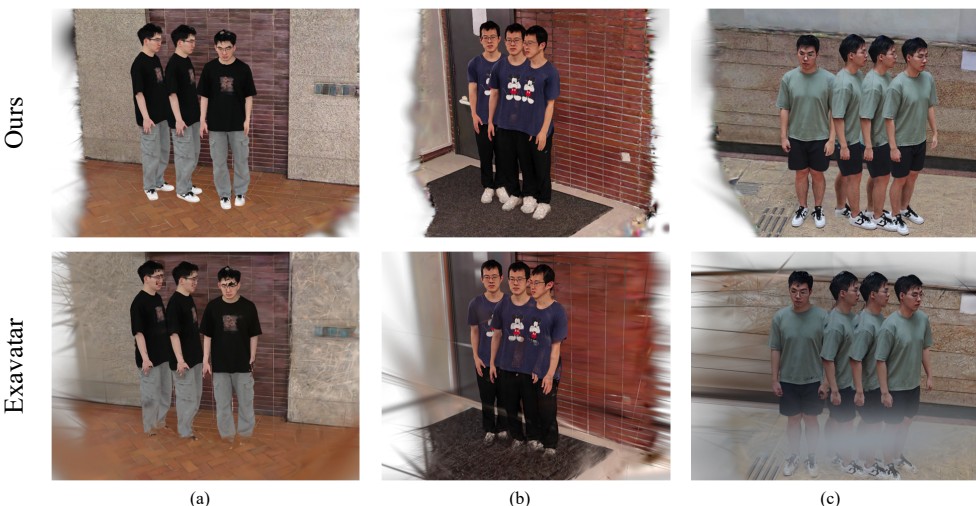

Figure 7: Qualitative 4D human-scene reconstruction results

Each acting video is approximately 15 to 25s long, with a frame rate of 10Hz. All of the frames are used for both training and testing. To quantitatively evaluate the model's ability of novel view synthesis, we recorded an additional evaluation video at a fixed camera pose while recording three of the acting video, providing the ground truth image at the novel view.

### 5.1.2 METRICS

We report quantitative results using three standard metrics at the evaluation view: Peak Signal-to-Noise Ratio(PSNR), Structure Similarity Index Measure(SSIM) and Learned Perceptual Image Patch and Similarity(LPIPS) Zhang et al. (2018). In addition, we introduced the Normal Mean Angle Error and the Normal Median Angle Error using the normal map obtained from sapiens Khirodkar et al. (2024) as the ground truth to quantitatively evaluate the geometry of the avatar. We also calculate the Mask IoU at the evaluation view to verify whether the avatar is localized in the correct 3D position and pose.

### 5.1.3 BASELINES

We compare our methods with the state-of-the-art 4D reconstruction methods: 4D Gaussian Wu et al. (2024a)(Template Free), HSR Xue et al. (2024) (Neural Rendering-based) and ExAvatar Moon et al. (2024a)(Gaussian Splatting-based). For a fair comparison, we disabled our camera refinement module, since both 4D Gaussian and Exavatar operate with fixed camera poses without optimization. We adopt baselines' own pre-process methods if provided, but ours if not.

### 5.2 EVALUATION ON THE DATASET

We evaluate 4D Gaussians, HSR, and Exavatar on three datasets that provide additional evaluation videos, and report rendering results from novel viewpoints.

As a hybrid representation combining 3D Gaussians and meshes, Exavatar can realistically capture facial expressions and body movements, leading to avatar reconstruction results comparable to ours. However, as shown in Fig. 7, when rendering multi-frame sequences of both the scene and humans from novel viewpoints, Exavatar exhibits noticeable artifacts. Specifically, opacity leakage occurs in the avatar, particularly in the face and upper body regions in scenes (a) and (b). In all tested scenes, Exavatar also suffers from interpenetration between the avatar and the static environment, due to the lack of joint optimization of geometry. Moreover, in scene (c), the reconstructed environment contains numerous large floaters. It is caused by the lack of visual features on the gray background wall. As shown in Fig. 8, HSR incorporates monocular depth and normal priors during optimization and continually adjusts their relative scales, which helps to avoid interpenetration artifacts. However,

Table 1: Quantitative comparison with baselins at evaluation view. Lower is better except PSNR(↑).

| Method | PSNR ↑ | D-SSIM ↓ | LPIPS ↓ |
|---|---|---|---|
| 4D Gaussian Wu et al. (2024a) | 16.72 | 0.62 | 0.55 |
| HSR Xue et al. (2024) | 20.46 | 0.51 | 0.53 |
| Exavatar Moon et al. (2024a) | 20.38 | 0.38 | 0.41 |
| **Ours** | **23.57** | **0.30** | **0.27** |

Table 2: Quantitative comparison on the Normal Angle Error of monocular 4D human-scene reconstruction.

| Method | Mean ↓ | Median ↓ |
|---|---|---|
| Exavatar | 41.34 | 31.94 |
| **Ours** | **23.64** | **13.54** |
| **Ours w/o** $\mathcal{L}_{\text{normal}}$ | 36.03 | 25.52 |

Table 3: Quantitative comparison on the Mask IoU of monocular 4D human-scene reconstruction.

| Method | Mask IoU ↑ |
|---|---|
| **Ours** | **0.91** |
| **Ours w/o** $\mathcal{L}_{\text{contact}}$ | 0.78 |

the reconstructed human motions are still significantly distorted, particularly in the second scene. In addition, the rendered scenes from HSR fail to capture the details of complex textures on the ground. In the first scene, HSR is also unable to reliably distinguish between static and dynamic human, resulting in floaters in background regions without avatar.

4D Gaussians, as a template-free approach, can faithfully reconstruct static scenes. Nevertheless, it fails to properly model human appearance and motion in a canonical space, leading to poor performance under novel view synthesis.

In quantitative evaluations, our approach achieves consistently superior performance over all three baselines across standard metrics, including PSNR, SSIM, and LPIPS, as shown in Table 1.

## 5.3 ABLATION STUDIES

As shown in Fig. 5, removing the avatar prior leads to noticeable color bleeding in unseen regions and erroneous SMPL-X parameter estimation of the hand. In addition, removing the scene prior results in sparse static scene point clouds and opacity leakage, especially when rendering from novel viewpoints. These observations confirm the importance of incorporating scene and human priors for complete 4D human-scene reconstruction.

Qualitatively, as presented in Fig. 6, the absence of normal supervision severely degrades the geometric quality of the avatar and incorporating the normal loss significantly enhances geometric reconstruction under novel view synthesis and reduces the Normal Angular Error, as illustrated in Table 2. Qualitatively, as shown in Fig.4, the absence of contact constraints results in interpenetration between the human and the scene. Quantitatively, incorporating the contact loss significantly improves Mask IoU under novel view synthesis, as shown in Table3, thereby ensuring a correct 3D spatial relationship between the human and the scene.

## 6 CONCLUSION

In conclusion, we propose a novel framework for complete 4D human–scene reconstruction from monocular videos, addressing the limitations of prior methods and enabling faithful novel-view synthesis. Our method jointly optimizes camera poses, human motion, and scene geometry by introducing a regularization strategy and combinig static 3D Gaussian scene representation with an animatable SMPL-based avatar. Experimental results demonstrate that our approach recovers time-varying geometry and appearance, achieving state-of-the-art performance and significantly outperforming existing monocular baselines.

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

## A THE REASON FOR NORMAL SUPERVISION

Common forms of geometric supervision include depth and normal supervision. This work has tried both of these two methods and found the normal supervision is more useful in the end. There are two common types of mono depth supervisions, one is depth ordering loss, the other is pixel-wise difference loss. In the context of human–scene reconstruction, this work find that while existing monocular depth estimation methods provide reliable relative depth ordering, it could provide such little benefit for SMPL-driven avatar optimization. Moreover, their per-pixel metric depth predictions remain unsatisfactory. The per-pixel depth supervision is so strict that this unsatisfactory prediction, combined with the additional inaccuracies introduced during scale alignment, can significantly undermine the effectiveness of geometry optimization. In contrast, the inferred normal maps are usually much more reliable and provide valuable guidance for improving the avatar's surface geometry.

In particular, even though color and normal rendering at the acting view of the avatar are reliable, there is no guarantee that the avatar is reconstructed with the correct spatial position in the scene. With the pinhole camera model, scaling the avatar by an arbitrary factor produces the same color and normal renderings in the camera coordinate system. To resolve the issue of scale ambiguity, it is essential to impose additional constraints that enforce contact between the avatar and the surrounding environment.

## B THE REASON FOR SCALE SUPERVISION

In particular, even though color and normal rendering at the acting view of the avatar are reliable, there is no guarantee that the avatar is reconstructed with the correct spatial position in the scene. With the pinhole camera model, scaling the avatar by an arbitrary factor produces the same color and normal renderings in the camera coordinate system. To resolve the issue of scale ambiguity, it is essential to impose additional constraints that enforce contact between the avatar and the surrounding environment.

## C THE USE OF LARGE LANGUAGE MODEL

In accordance with the Policies on Large Language Model Usage at ICLR 2026, we hereby disclose that an LLM was utilized solely for improving the grammar and wording of this paper. The LLM was employed to refine the clarity and readability of the text, ensuring that the manuscript adheres to

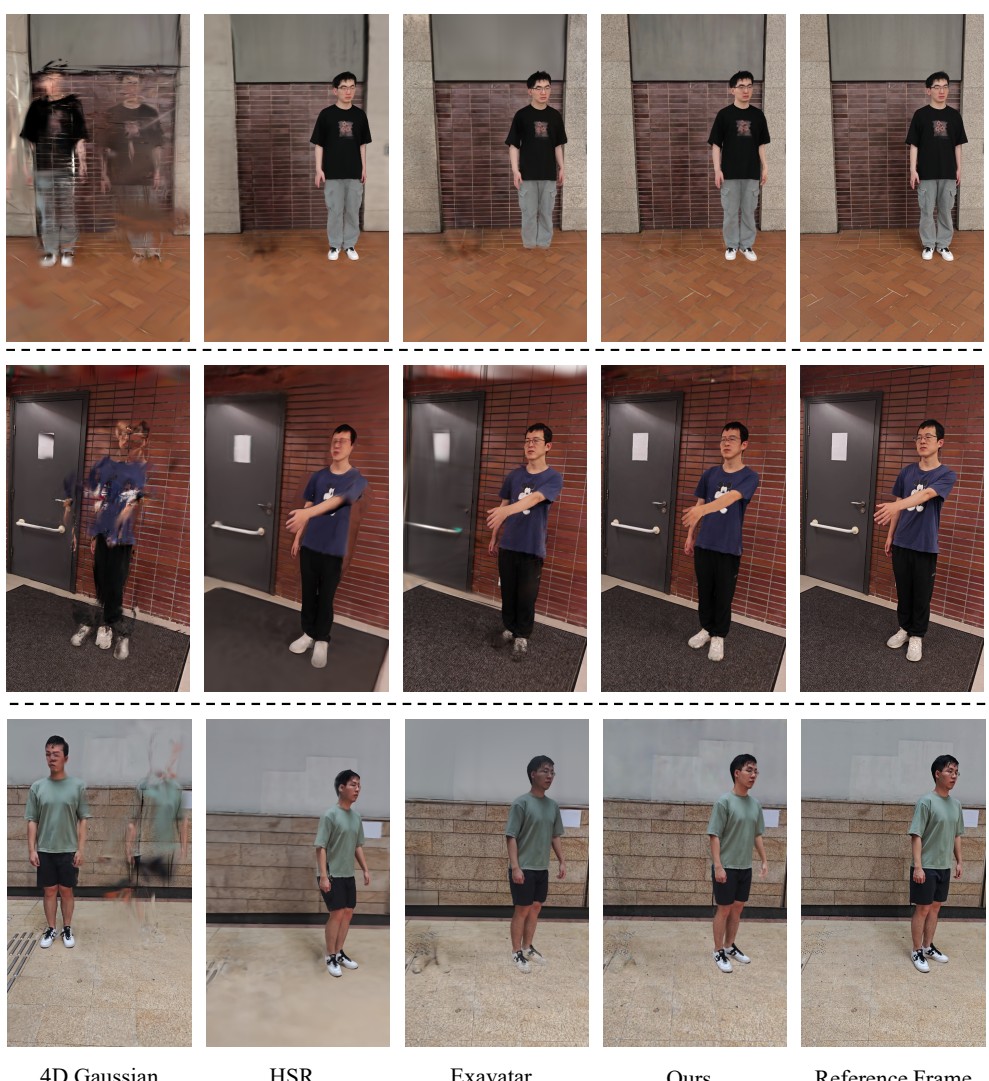

| 4D Gaussian | HSR | Exavatar | Ours | Reference Frame |

Figure 8: Qualitative Comparison at novel evaluation view with the baselines

high linguistic standards. We, the authors, take full responsibility for the content of this submission, ensuring that all claims, data, and results are accurate and free from misrepresentation.

