# OpenReview forum: "Prior-based 4D Human-Scene Reconstruction from Monocular Videos"
_ICLR.cc/2026/Conference — ICLR 2026 Conference Withdrawn Submission_

### Official Review · Reviewer_mJFr · 2025-10-30

**Soundness:** 2
**Presentation:** 2
**Contribution:** 1
**Rating:** 2
**Confidence:** 4

**Summary:**

This paper proposes a 4D human-scene reconstruction method that integrates a 3DGS scene prior and a SMPL-based Gaussian avatar prior, and jointly optimizes these priors, achieving realistic rendering results

**Strengths:**

- Integrates human and scene reconstruction.
- Provides clear qualitative comparisons and demo videos.

**Weaknesses:**

- Lack of novelty and insufficient contribution. The method mainly combines existing 3DGS-based scene reconstruction and avatar models with only minor optimization refinements. The avatar deformation design appears identical to ExAvatar, offering no clear methodological innovation.

- Unfair comparison. The proposed method uses three videos per subject for reconstruction, whereas ExAvatar operates on a single monocular video.

- Invalid evaluation protocol. As stated in L398, "All of the frames are used for both training and testing," which is not a reasonable practice. The absence of a proper train/test split undermines the validity of the reported metrics.

- Misleading use of the term "prior". The "scene prior" and "avatar prior" are not true learned priors but pre-reconstructed assets obtained from other models or separate videos. This design relies on external pre-fitted results rather than an intrinsic modeling prior.

- Poor organization and unclear notation. Several variables and equations are undefined or poorly explained. For example, in L250, the meanings of $E$ and $P_c$ are not clarified. The regularization loss is vaguely described and lacks a supporting mathematical formulation. Training details are missing, and typos (e.g., inconsistent use of "Exavatar" vs. "ExAvatar") further affect the paper’s clarity

**Questions:**

1. What's the difference between the proposed avatar deformation and ExAvatar's? Is this deformation sufficient to handle complex human–scene interactions, especially when obstacles or physical contacts occur?
2. Please provide explicit explanations for all notations used.
3. The paper lacks important implementation details such as learning rates, loss weights, training iterations, and optimization settings. How efficient is your method compared with existing approaches in terms of training time and computational cost?

---

### Official Review · Reviewer_7RWz · 2025-10-30

**Soundness:** 2
**Presentation:** 2
**Contribution:** 2
**Rating:** 4
**Confidence:** 4

**Summary:**

This paper presents a novel framework for 4D human-scene reconstruction from a single monocular video, aiming to overcome the limitations of current methods that either reconstruct the human or the scene in isolation or require complex multi-view camera setups. The proposed approach jointly reconstructs the dynamic human and the static 3D environment by integrating a static 3D Gaussian Splatting scene prior with an animatable, SMPL-based Gaussian human prior. By simultaneously optimizing for camera pose, human motion, and the parameters of both priors, the system effectively separates dynamic and static elements, achieving realistic human-scene interactions and smooth temporal coherence. The method addresses key challenges such as scale ambiguity between the human and scene, non-rigid clothing deformation, and appearance discrepancies, demonstrating state-of-the-art performance on a self-collected dataset.

**Strengths:**

- The method enables joint reconstruction of humans and their surrounding 3D environments from a single monocular video, removing the need for multi-view setups and making real-world deployment more practical.
- It integrates a static 3D Gaussian Splatting scene prior with an animatable SMPL-based Gaussian human prior, effectively separating static and dynamic components while preserving geometric detail.
- By jointly optimizing camera poses, human motion, and Gaussian parameters, the system achieves smooth temporal coherence and realistic human–scene interactions.

**Weaknesses:**

- Lack of evaluation on any publicly available benchmark. The method is only validated on a self-collected dataset, making it difficult to assess generalization or compare fairly with existing approaches.
- The proposed method relies on two separate scanned videos—one for the human and one for the static scene—before performing the joint reconstruction. This assumption severely limits the method’s practicality and reproducibility. In real-world monocular scenarios, obtaining clean and aligned human and scene scans is often infeasible. Moreover, all sequences are self-captured, and the paper provides no information about how these scans are temporally or geometrically synchronized. As a result, the method’s applicability to in-the-wild videos remains unclear, and the evaluation cannot demonstrate generalization beyond the authors’ controlled recording setup.
-  Using ExAvatar both as a component and as a baseline conflates ablation with external comparison, making the evaluation methodologically unsound.

**Questions:**

- The details of how to benchmark the ExAvatar.
- How the scene scan is constructed? Does the scene scan contains the human?

---

### Official Review · Reviewer_Z7uV · 2025-10-31

**Soundness:** 2
**Presentation:** 2
**Contribution:** 2
**Rating:** 2
**Confidence:** 4

**Summary:**

This paper aims to reconstruct complete 4D scenes involving human-scene interactions from monocular video. Three videos are taken as input: a monocular scanning video of the scene, a monocular scanning video of the human, and an acting video for joint human-scene reconstruction. The scene geometry is recovered via 3DGS optimization with Depth Anything V2 priors. The human avatar is recovered using off-the-shelf Gaussian avatar reconstruction ExAvatar. Then, the human and scene are jointly optimized to minimize a ground-plane contact loss, photometric loss, and normal loss.

**Strengths:**

- The authors address a challenging yet important task of jointly reconstructing scenes and 4D humans from monocular video, by combining feed-forward priors (e.g. DAv2, Sapiens normals, HMR) with Gaussian optimization.

**Weaknesses:**

- The paper provides limited quantitative and qualitative results. The main results shown are on a single self-collected dataset, which has limited complexity in scene and human motions (e.g. all three supplementary videos have a mostly static camera with the same action of turning and walking). Compared to existing published work for dense human-scene reconstruction (e.g. https://eth-ait.github.io/ODHSR/), the applicability of the method on truly in-the-wild videos seems limited.
- Sec. 4.1 claims that the method requires three videos as input (a human scanning video, a scene scanning video, and a joint human-scene acting video). The need for human scanning video and scene scanning video is not assumed by existing work which operates directly on monocular videos (e.g. ODHSR, Shape of Motion), limiting the applicability of the method on unconstrained smartphone videos (as proposed in the introduction).
- Quantitative and qualitative evaluation on at least one standard dataset (e.g. the SHSD proposed by HSR paper) would strengthen the results of the paper. There are many standard human-scene interaction datasets (EMDB, PROX, NeuMan) and multi-view studio captures permitting novel-view evaluation (Panoptic Studio, DNA-Rendering). Due to the need for human scanning and scene scanning videos as input, the paper claims that these datasets are not informative enough (personally I feel this is a limitation).
- The method assumes constant foot contact with a ground plane, which is not the case even in simple human-scene interaction activites such as walking.

**Questions:**

- To my understanding, the paper uses three stages of optimization (static scene reconstruction, human reconstruction, and joint optimization). What are the losses used during each optimization stage? Do the photometric loss, normal loss, and contact loss only apply to the joint optimization stage?

---

### Official Review · Reviewer_o48N · 2025-11-01

**Soundness:** 3
**Presentation:** 3
**Contribution:** 3
**Rating:** 6
**Confidence:** 3

**Summary:**

This paper studies how to reconstruct both a moving human and the surrounding scene from a single camera video. The authors use two priors. One for the static scene with 3D Gaussian Splatting. One for the human body based on SMPL with 3D Gaussians. They train the system by optimizing camera pose, human motion, and both priors together. They use normal, contact, and photometric losses to make the result more stable. They also collect a small dataset of human and scene videos for testing. The results look good and beat several strong baselines like ExAvatar and HSR. It gives around 23 dB PSNR on novel view rendering. The method works well visually and shows clear improvement.

**Strengths:**

* Clear motivation and solid method.
* The use of normal and contact loss helps with geometry.
* Good results compared to strong baselines.
* Figures show visible improvement.
* Ablation studies are complete and easy to understand.

**Weaknesses:**

* The first weakness is the small dataset. The authors only use a few self-captured videos and scans. This makes the evaluation narrow and limits how much we can trust the generalization. It would be better to test on public datasets or unseen environments.
* I believe the work is missing discussion of runtime and efficiency. There is no information about how long optimization takes or how big the model is. It is important because Gaussian-based methods can be heavy, and readers want to know if this can run in real time or not.
* Lack of large-scale comparisons is also considerable in my opinion. The paper only compares to a few baselines. It would be more convincing to include more recent dynamic Gaussian or NeRF-based models to show stronger evidence.
* Another weakness is that the method might not handle diverse motions or multiple people. It is unclear if it can generalize beyond single-person scenes. The framework seems limited to simple interactions.
* Finally, some parts of the method section are hard to follow, especially the optimization details. Variables appear without clear definition, and equations are sometimes dense. It would help to explain the intuition behind the math more.

**Questions:**

1. How long does it take to optimize one sequence? Can it be used for real-time or interactive applications, or is it mostly offline?

2. How general is the method? Have you tried videos with different people, outfits, or background settings? Does the model still perform well?

3. You use your own dataset. Would the method still work on standard benchmarks such as Human3.6M or ZJU-Mocap?

4. How sensitive is your approach to errors in SMPL fitting? If the initial pose or scale is wrong, does the optimization recover it?

5. What would happen if the priors are noisy or low-quality? Does the system rely heavily on accurate scan priors, or can it still work with rough inputs?

6. Could the framework be extended to handle more than one person or simple object interactions?

---

### Note · Authors · 2026-03-07

I have read and agree with the venue's withdrawal policy on behalf of myself and my co-authors.

---

### Meta-Review · Area_Chair_Tucq · 2025-12-29

**Summary:**

This paper focuses on the problem of 4D human avatar reconstruction from monocular video. The proposed method is straightforward and does not present many novel ideas. Also, the proposed method relies on two separate scanned videos, one for the human and one for the static scene, before performing the joint reconstruction, which is unfair when compared to others. The paper only conducts experiments on self-collected and small-scale datasets and lacks evaluation on public benchmarks. Finally, the author didn't submit a rebuttal to address the reviewers' concerns.

**Reviewer Concerns:**

The author didn't submit any feedback or rebuttal. The main concerns are summarized in the Summary section above.

**Reviewer Scores:**

The reviewers didn't change the score or provide any evidence to change it.

---

### Decision · Program_Chairs · 2026-01-26

Reject